# DFT Study of WS_2_-Based Nanotubes Electronic Properties under Torsion Deformations

**DOI:** 10.3390/nano13192699

**Published:** 2023-10-04

**Authors:** Anton V. Domnin, Ilia E. Mikhailov, Robert A. Evarestov

**Affiliations:** Quantum Chemistry Department, St. Petersburg State University, 7/9 Universitetskaya nab., 199034 St. Petersburg, Russia; st087328@student.spbu.ru (I.E.M.); r.evarestov@spbu.ru (R.A.E.)

**Keywords:** density functional theory (DFT), line groups theory, nanotubes, torsion deformation, tungsten disulfide

## Abstract

In this study, the influence of torsional deformations on the properties of chiral WS_2_-based nanotubes was investigated. All calculations presented in this study were performed using the density functional theory (DFT) and atomic gaussian type orbitals basis set. Nanotubes with chirality indices (8, 2), (12, 3), (24, 6) and (36, 9) corresponding to diameters of 10.68 Å, 14.90 Å, 28.26 Å and 41.90 Å, respectively, are examined. Our results reveal that for nanotubes with smaller diameters, the structure obtained through rolling from a slab is not optimal and undergoes spontaneous deformation. Furthermore, this study demonstrates that the nanotube torsion deformation leads to a reduction in the band gap. This observation suggests the potential for utilizing such torsional deformations to enhance the photocatalytic activity of the nanotubes.

## 1. Introduction

Tungsten disulfide (WS_2_) is a promising material for nanotechnology due to its layer structure and hexagonal lattice, which allow the formation of nanosheets and nanotubes. The first WS_2_ nanotubes were synthesized in 1992 by Tenne [1], and since then, extensive research has been conducted to explore their unique properties and applications. Numerous scientific articles are devoted to experimental studies on WS_2_ nanotubes. Modern synthesis methods involve a heterogeneous reaction between powdered WO_3−x_ and a gaseous mixture of H_2_S and H_2_/N_2_ at temperatures ranging from 800–950 °C. Using this method, nanotubes with lengths of 1–10 µm and diameters of 20–150 nm can be obtained [2]. In 2017, Chen et al. [3] successfully measured the precise diameters and chirality indices by analyzing electron diffraction patterns of the corresponding nanotubes.

Moreover, WS_2_-based nanotubes have been comprehensively investigated theoretically using quantum mechanics, molecular dynamics, and molecular mechanics methods. In [4,5], the electronic structure of WS_2_ single-walled (SWNT) and multi-walled nanotubes with various chirality indices was calculated using DFT-LCAO and DFT-LACW methods. This allows obtaining the desired value of the band gap by varying the diameter of NTs [4]. It was also shown that dopants and structural defects can also influence the electronic properties of NT [5]. Studies [6,7] indicated that one-dimensional transition metal structures, like MoSSe-WSSe double-walled Janus nanotubes and MoS_2_ NT/PbS_x_Se_1−x_ quantum dots heterostructures, demonstrated the possibility of ultrafast charge transfer, which is remarkable for their application as solar cells. Alternatively, one can explore the mechanical deformation of nanotubes with fixed chirality, rather than studying nanotubes with different chirality indices.

The geometry of nanotubes can be completely characterized by chirality indices (*n*_1_, *n*_2_). These indices provide information about the nanotubes’ symmetry, length of the translation vector, number of atoms in the unit cell, and diameter [8,9,10]. Nanotubes with (*n*, *n*) and (*n*, 0) indices are achiral, possessing inversion symmetry, while other nanotubes are chiral, exhibiting left and right screw axes [10]. Different chirality indices correspond to nanotubes with distinct electronic properties, including the electronic band gap energy.

It is well-known that the electronic properties of nanotubes, such as the band gap, the positions of the valence band edge ($E_{VB}$) and conduction band edge ($E_{CB}$), depend on the nanotube’s deformation. Several studies have been dedicated to the investigation of the axial deformation of MoS_2_ and WS_2_ nanotubes [11,12,13]. These studies have demonstrated that MoS_2_ nanotubes can be stretched up to 16%, and WS_2_ nanotubes up to 12% before rupture. The electronic band structure of these nanotubes can be tuned by applying tensile stress, making them potential candidates for applications in nanoelectronics as switches and piezoresistive strain sensors.

Another type of deformation is torsional deformation, which has been studied theoretically in carbon and silicon nanotubes [14,15,16]. In particular, D’yachkov investigated [13] carbon and silicon nanotubes using the linearized augmented cylindrical waves (LACW) method. He demonstrated that adjusting the torsion angle *ω* affects the band gap as well as the positions of the valence band top ($E_{VB}$) and conduction band bottom ($E_{CB}$). The dependence of the band structure on *ω* differs between chiral and achiral nanotubes. However, D’yachkov’s study limited the range of torsion angle ω to $\left[-2^{\circ },2^{\circ }\right]$ degrees to prevent potential rupture of the nanotubes. It is important to note that within this approach, quantum chemical calculations are performed without geometry optimization, which can potentially impact the accuracy of the obtained results.

In [17], torsional deformations of MoS_2_ nanotubes were investigated using molecular dynamics, and focusing solely on mechanical properties. In a subsequent work [18], ab initio DFT calculations were performed on TMD nanotubes with different metals, but only achiral nanotubes were studied, and only axial deformations were considered. Therefore, there is a need for detailed first-principle calculations and the characterization of chiral WS_2_-based NTs, taking into account the dependence of the electronic properties on the torsion angle.

Several studies have been conducted to experimentally investigate torsion deformations of WS_2_-based nanotubes. In [19,20], a method is described where a WS_2_ nanotube is fixed at its edges, and an AFM (Atomic Force Microscopy) needle pushes a lever fixed at the middle of the NT, causing torsional deformation of the structure. This technique ensures accurate measurement of mechanical and electrical properties, such as the shear modulus and conductivity of the material. In addition, theoretical studies have been conducted using semi-empirical DFTB (Density Functional based Tight Binding) method to examine the band structure changes of achiral WS_2_-based nanotubes under torsional deformations [21]. The results have demonstrated a correlation between experimental and theoretical data, indicating that conductivity increases as the torsion angle increases. However, since most synthesized nanotubes are chiral, it would be more accurate to study this type of nanotubes.

This study focused on the effect of torsional deformations on the properties of WS_2_ nanotubes with chirality indices *n*(4, 1) where *n* = 2, 3, 6, 9 (see Figure 1). Torsion deformations were modeled using the line groups theory approach. By taking symmetry into account at each stage of the calculations, the complexity of the quantum chemical calculations was significantly reduced. We obtained dependencies of the relative energies, the electronic band gap energies, and the energies of the top of the valence band and the bottom of the conduction band on the torsion angle. This technique has already been used to study a variety of carbon nanostructures, including polytwistane [22], nanohelicenes [23], and polyacetylene [24]. However, distinct from [17,18,19], ab initio methods are explored for the first time in this study to investigate the properties of nanotubes as a function of torsion angle.

Currently, methods for determining the chirality indices of experimentally synthesized nanotubes are known [3,25]. Furthermore, the recent work by An et al. [26] proposed a method for synthesizing nanotubes with a fixed chiral angle. However, multi-walled nanotubes with large diameters (approximately 100–200 Å) are typically synthesized. One approach to modeling such nanotubes is using molecular mechanics [27]; however, even this cannot provide an accurate representation of experimentally obtained nanotubes. Further research is required to develop ab initio simulation techniques capable of accurate reproducing nanotubes with diameters closely matching those observed in experiments.

## 2. Computational Details and Elements of Line Groups Theory

### 2.1. Computational Details

Ab initio calculations in this work were performed with CRYSTAL17 [28,29] program using a hybrid version of DFT exchange–correlation functional HSE06. The relativistic effective core pseudopotentials and the corresponding basis sets were taken the same as in [4]. To provide a balanced summation in both direct and reciprocal lattices, the reciprocal space integration was performed by sampling the Brillouin zone (BZ) with the 18 × 1 × 1 Monkhorst–Pack mesh [30], which results in a total of 10 k-points evenly distributed over BZ. Calculations were considered as converged only when the total energy differs by less than 10^−9^ a.u. in two successive cycles of the self-consistent field procedure. Strict accuracy criteria (10^−8^ 10^−8^ 10^−8^ 10^−8^ 10^−18^) were used in the calculations of the Coulomb and exchange series. These tolerances mean that during the direct lattice summations, the one-electron integrals and two-electron Coulomb integrals less than 10^−8^ are estimated by the multipolar expansion, and two-electron exchange integrals less than 10^−16^ are ignored. For more comprehensive information, please refer to the CRYSTAL17 User’s Manual [31]. Both the translational period and atomic positions have been optimized. The gradient threshold was set equal to 0.00005 Ha·Bohr^−1^. Convergence criterion on the atomic displacement was set equal to 0.0001 Bohr.

By incorporating roto-helical symmetry throughout the calculation, both computational and memory costs were significantly reduced. To validate the computational scheme, atomic and electronic structure calculations were performed for 2H-WS_2_ and a single 0001 monolayer composed of three atomic planes (S-W-S). Experimental measurements revealed a fundamental band gap of 2.4 eV for the WS_2_(0001) nanosheet with 1 monolayer thickness when supported by Si [32] or α-Al_2_O_3_ [33] substrates, which is considerably larger than the experimental band gap of bulk WS_2_ (1.4 eV).

The results obtained by us from the quantum chemical simulations demonstrate good agreement with the experimental crystal lattice parameters and reasonable agreement with the electronic band gap, see Table 1.

### 2.2. Application of Line Group Theory to Nanotubes with Hexagonal Morphology

Let us consider a brief overview of the theoretical modeling of nanotubes with hexagonal morphology using the theory of line groups. For more comprehensive information, please refer to the monographs [10,36] and articles [8,9,37]. Article [38] is devoted to the study of the symmetry of nanotubes based on transition metal dichalcogenides.

Structures of tubes with a hexagonal morphology, obtained by rolling a hexagonal layer ($a=\left|{\vec{a}}_{1}\right|=\left|{\vec{a}}_{2}\right|\;$; $\alpha =60^{\circ }$), are defined using orthogonal vectors $\vec{R}\;\cdot \;\vec{T}=0$ [10]. Here, $\vec{T}$ is a translation vector and $\vec{R}$ is called a chiral vector, which becomes a circumference of nanotube. Furthermore, $\vec{R}=n_{1}{\vec{a}}_{1}+n_{2}{\vec{a}}_{2}$ where $n_{1}$ and $n_{2}$—are the chirality indices of the nanotube. This can also be rewritten as $n\left({\tilde{n}}_{1},{\tilde{n}}_{2}\right)$, where n is the greatest common divisor for $n_{1}$ and $n_{2}$. The chirality indices also determine the components of the translation vector $\vec{T}=t_{1}{\vec{a}}_{1}\;+\;t_{2}{\vec{a}}_{2}$ by the formula:(1)$$\frac{t_{1}}{t_{2}}=-\frac{2{\tilde{n}}_{2}+{\tilde{n}}_{1}}{2{\tilde{n}}_{1}+{\tilde{n}}_{2}}$$

Another important characteristic of nanotube structure is the spiral vector $\vec{H}=h_{1}{\vec{a}}_{1}+h_{2}{\vec{a}}_{2}$. The indices $h_{1}$ and $h_{2}$ can be determined from the formula:(2)$${\tilde{n}}_{1}h_{2}-{\tilde{n}}_{2}h_{1}=1$$

Figure 2 shows the chirality vectors $\vec{R}$ for nanotubes with the different chirality indices and hexagonal morphology. It is seen that the chirality vectors belong to one ray if $\left({\tilde{n}}_{1},{\tilde{n}}_{2}\right)$ is fixed and n = 1, 2, 3, …

When folding a WS_2_ layer into a chiral nanotube, the layer’s symmetry group is replaced by the line group $L=T_{q}^{r}C_{n}$, which belongs to the first family of line groups. $T_{q}^{r}$ is referred to as the generalized translation group and is represented by the operation $Z=T\left(C_{Q}|f\right)$, where T is nanotube translational group if Q is rational, i.e., $Q=\frac{q}{r}=\frac{n\tilde{q}}{r}\geq 1$ represents the order of the helical axis (rotation angle of the helical axis $\varphi =\frac{360{}^{\circ}}{Q}$), and $f=\frac{t\;}{\tilde{q}}$ is a partial translation ($t=\left|\vec{T}\right|$). $C_{n}$ represents the symmetry group of the monomer, consisting of 3·n atoms. This form of representing line group is known as the “polymer” or “helical” factorization. Conversion to the crystallographic notation is based on the calculation of the *p* index, required for this type of notation ***L*** = *Lq_p_* [36]. All parameters of the symmetry group can be easily determined by knowing the vectors $\vec{R}$, $\vec{T}$, and $\vec{H}$:(3)$$q=n\tilde{q}=n\left|\begin{array}{cc} {\tilde{n}}_{1} & {\tilde{n}}_{2}\\ t_{1} & t_{2} \end{array}\right|$$
(4)$$r=h_{1}t_{2}-h_{2}t_{1}$$
(5)rp=lq+n
(6)$$r\tilde{p}=l\tilde{q}+1$$

Here, *l* is the positive integer or zero.

For instance, let us consider a nanotube with chirality indices (10, 5). By finding the greatest common divisor (GCD) between $n_{1}$ and $n_{2}$, one can simplify the notation to 5(2, 1), where n=5. Using Formulas (1) and (2), one can determine the components of the vectors $\vec{T}$ and $\vec{H}$: $t_{1}=-4$, $t_{2}=5$, $h_{1}=h_{2}=1$. With this information, one can calculate the parameters for the symmetry group of the selected nanotube using Formulas (3)–(6). It can be found that $q=n\tilde{q}=70$, r=9, $Q=\frac{70}{9}$, and p=55. Therefore, the symmetry group of the (10, 5) nanotube in the helical factorization is written as $T_{70}^{9}C_{5}=ZP$, where $Z\equiv T_{70}^{9}=T\left(C_{70}^{9}|f\right)$, and $P\equiv C_{5}$.

### 2.3. Torsion Deformation of NTs with Hexagonal Morphology

From a practical standpoint, it is convenient to introduce the variable $\tilde{Q}=\frac{\tilde{q}}{r}$, which determines the parameters of the helical axis, and the angle $\tilde{\varphi }=\frac{360^{\circ }}{\tilde{Q}}$. The use of $\tilde{Q}$ is advantageous because, for any set of tubes specified by the formula $n\left({\tilde{n}}_{1},{\tilde{n}}_{2}\right)$, the value of $\tilde{Q}$ remains unchanged even if Q undergoes multiple changes.

In order to identify groups that describe deformed nanotube (NT) structures, it is necessary to find for each of them ${\tilde{Q}}_{x}=\frac{{\tilde{q}}_{x}}{r_{x}}$ such that the torsion angle $\omega ={\tilde{\varphi }}_{x}-\tilde{\varphi }$ falls within a certain range. This range is determined by the specific structure and composition of the nanotube. Table 2 presents the torsion angles and their corresponding symmetry groups, indicated in both crystallographic ($L{n\cdot \tilde{q}}_{n\cdot \tilde{p}}$) and helical notations ($T_{q}^{r}C_{n}$).

By utilizing the Formulas (3) and (4), the value of $\tilde{Q}$ for the initial nanotube can be easily determined. With the aid of an enumeration method, it is possible to identify a set of $\tilde{q}$ and *r* values for which the torsion angle ω falls within the specified range. It is important to impose a maximum limit on the $\tilde{q}$ value to avoid excessive computational complexity. Increasing $\tilde{q}<$ ${\tilde{q}}_{max}$ leads to a corresponding increase in the number of formula units in the elementary cell according to the formula $q=n\tilde{q}$, thereby complicating the quantum chemical calculations.

## 3. Results and Discussion

### 3.1. Results of Calculations for Non-Deformed Nanotubes

In the present paper, we consider torsion deformation of WS_2_-based chiral nanotubes n(4, 1) for n=2, 3, 6, 9. The results of our calculations are provided in Table 3. The crystallographic factorization ${Lq}_{p}$ of the corresponding line groups is provided in the second line of Table 3.

The input values of NT diameters $D=\frac{a}{\pi }n\sqrt{{\tilde{n}}_{1}^{2}+{\tilde{n}}_{1}{\tilde{n}}_{2}+{\tilde{n}}_{2}^{2}}$ are increasing from 9.22 Å to 41.49 Å. It is seen that due to the geometry optimization the change of the diameter ΔD decreases with the input diameter increasing as for the monolayer, this change becomes formally zero.

For all NTs in consideration, the translation vector $\vec{T}$ length t is the same as it is defined by the Z subgroup of line group L in the polymer factorization L=ZP ($P=C_{2},C_{3},C_{6},C_{9}$; $Z=T{\left(C_{q}^{11}|f\right)}^{l}$; l=0, 1, …q−1, t=14f). It is seen from Table 3 that the translation vector length t changes due to the geometry optimization decreasing with NT diameter increasing, and moves to value 0.01 Å—the difference of the bulk and monolayer hexagonal lattice vector length *a*.

Table 3 shows that the primitive unit cell energy E value per formula unit decreases with NT diameter increasing, thus moving to that for monolayer.

The formation $E_{form}$ and strain $E_{str}$ energies of NTs are defined by relations:(7)$$E_{form}=\frac{E}{N}-\frac{E_{bulk}}{N_{bulk}}$$
(8)$$E_{str}=\frac{E}{N}-\frac{E_{mono}}{N_{mono}}$$
where *N*, $N_{bulk}$ and $N_{mono}$ are the numbers of formula units in the primitive cell for nanotube, bulk crystal, and monolayer. The corresponding energies per primitive unit cell are *E*, $E_{bulk}$ and $E_{mono}$. It is evident that the difference ($E_{form}$ − $E_{str}$) = 22.56 kJ/mol is the same for all the nanotubes as it is the monolayer formation energy from bulk crystal.

Table 3 shows the forbidden energy gap (*E_gap_*) dependence on the nanotube diameter. As the energy gap for bulk is smaller than that for the slab (1.68 eV instead 2.53 eV), the band gap enlarges as the diameter of the nanotube (NT) increases. Moreover, it shifts toward the value of the band gap observed in a monolayer configuration. In the latter two lines of Table 3, we provide the one-electron energies $E_{v}$ and $E_{c}$ for the top of the valence band and the bottom of the conduction band, respectively. It is known that visible-light-driven water-splitting catalysts should meet following energy requirements: $E_{v}<-5.67$ eV and $E_{c}>-4.44$ eV (values of oxidation and reduction potentials of water). Our results show that this condition is fulfilled only for NTs (24, 6) and (36, 9). This conclusion is in agreement with that made in [4]: NTs with a diameter larger than 19 Å show the potential ability to serve as photocatalysts for water-splitting reactions.

### 3.2. Energy Minima and Atomic Structures of Torsional Deformed Nanotubes

For each nanotube analyzed, the impact of torsional deformation was studied within the range of $\omega \in \left[-2.857^{\circ },3.025^{\circ }\right]$. This specific torsion angle range was selected because it allows obtaining a single energy minimum for each nanotube in consideration. Further increasing the absolute value of torsion angle beyond this range would lead to an increase in the relative energy, indicating structural damage of the nanotube. The atomic structure of deformed NTs was optimized for each chosen torsion angle (see Table 2), and subsequent calculations were performed to determine NTs properties.

Figure 3 shows the dependence of the relative torsion energy per formula unit, which was calculated using the following formula:(9)$$\Delta E_{rel}\left(\omega \right)=\frac{E\left(\omega \right)}{q_{\omega }}-\frac{E\left(0\right)}{q_{0}}$$
where relative energy difference, denoted as $\Delta E_{rel}\left(\omega \right)$, is calculated as the difference between the total electronic energies of the original nanotube structure, $E\left(0\right),$ and the deformed nanotube structure with a torsion angle ω, $E\left(\omega \right).$ The number of formula units in the original nanotube is represented in (9) by $q_{0}$, while $q_{\omega }$ corresponds to the number of formula units in the nanotube structure with the torsion angle ω.

Figure 3 shows the dependence $\Delta E_{rel}$ on torsion angle. It is seen that when the nanotube diameter decreases, the position of the energy minimum deviates more significantly from ω=0. Furthermore, the preferred torsion direction varies based on the nanotube’s diameter. In the case of the smallest nanotube (8, 2), torsion towards positive ω is less favorable compared to larger diameter NTs. Conversely, within the range of negative ω, torsion of the largest nanotube is less preferable.

In Table 4, the characteristics of the energy minima obtained through $\Delta E_{rel}\left(\omega \right)$ interpolation are presented. An observed trend indicates that with an increase in nanotube diameter, the band gap increases while the shear modulus decreases. Additionally, the energy minimum shifts towards that of the undeformed nanotube structure.

Figure 4 illustrates the dependence on the torsion angle of structural parameters of the investigated nanotubes. In this context, an increase in the parameter partial (Figure 4a) translation $f\;=\;\frac{t}{\tilde{q}}$ corresponds to the stretching of the nanotube. As the nanotube diameter increases, the partial translation demonstrates an upward trend and moves to a constant value. Overall, the dependence of the partial translation on torsion angle remains, approximately, the same across all nanotubes; however, for the smallest nanotube (8, 2), the increase becomes slightly stronger when undergoing torsion within the range of positive torsion angles.

It is evident from Figure 4b that the diameter of the examined nanotubes displays slight fluctuations within the torsion angle range of $\left[-2^{\circ },2^{\circ }\right]$. Across all of the nanotubes assessed, the most significant deformation occurs at negative torsion angles, particularly with the nanotube (36, 9) displaying the most significant diameter changes. A comparable pattern is noticeable when deforming moves into the positive torsion angle region, however, the change in diameter is notably smaller.

Notably, it can be observed that the physical properties of nanotubes undergo more essential changes upon torsion as the initial nanotube diameter increases. The asymmetric band gap values relative to the torsion angle ω=0 are a consequence of the chiral nature of the nanotubes under investigation.

Figure 5 shows that the band gap exhibits slight variations for nanotubes with the smallest diameter (8, 2) and (12, 3) within the range of torsional deformation, with changes of no more than 0.15 eV and 0.25 eV, respectively.

For the nanotubes (24, 6) and (36, 9), the band gap decreases with torsional deformation, a trend that aligns with the findings of previous studies [19,21], demonstrating an increase in the electrical conductivity of WS_2_-based nanotubes due to torsion-induced deformation. While the observed trends in band gap for (24, 6) and (36, 9) are comparable, it is reasonable to hypothesize that nanotubes (NTs) with larger diameters exhibit similar tendencies in electronic properties.

Figure 6 illustrates the dependencies of the energies at the top of the valence band and the bottom of the conduction band for the nanotubes under investigation. It is evident that nanotubes (24, 6) and (36, 9) show potential applicability for water splitting, aligning with the established notion that nanotubes with smaller diameters are less suitable for this purpose.

Furthermore, it is apparent that as the diameter increases, the contribution of the top of the valence band to the change in $E_{gap}$ diminishes. Additionally, significant torsional deformation results in $E_{c}$ becoming lower than the reduction potentials of water.

Table 5 provides data on the behavior of the band gap under torsional deformations. Notably, for nanotube (8, 2), there is a transition from an indirect to a direct band gap in one case, whereas no similar transition occurs for the other nanotubes examined. Currently, we are unable to determine if this change is specific to nanotubes with small diameters or if it can be observed with stronger torsion. Nevertheless, this phenomenon warrants further investigation.

Based on the obtained data, it is evident that the nanotubes under investigation are not suitable for photocatalytic water-splitting, as they exhibit indirect bandgap semiconductor behavior.

## 4. Conclusions

In this study, we investigated the impact of torsional deformation on the properties of WS_2_-based chiral nanotubes using first principles analysis. We utilized line groups theory to reduce computational costs and applied density functional theory (DFT) for quantum chemical calculations.

Our findings reveal that nanotubes with smaller diameters exhibit a more significant deviation in the position of the energy minimum on the torsion curve. In this context, the smaller the diameter, the more substantial the deviation. A decrease in diameter also causes diminishing of relative diameter changes with partial translation increases, aligning with the nanotube’s stretching.

Additionally, our study shows that torsion has a minor impact on nanotube diameter within the range of −2° to 2°, but leads to more significant changes under stronger torsion, especially in the case of the nanotube (36, 9).

We observed that torsional deformation has a limited effect on the band gap in nanotubes with small diameters but becomes more noticeable as the nanotube diameter increases, aligning with previous research [21]. This property holds promise for applications in nanoelectronics based on WS_2_ nanotubes.

The study showed that the studied nanotubes cannot be used for photocatalytic water-splitting due to the indirect band gap. However, in the case of tube (8, 2), strong torsional deformation changed the band gap from indirect to direct, a phenomenon that deserves further study. Plots of band zones can be found in the Appendix A, specifically in Appendix A.

## Figures and Tables

**Figure 1 nanomaterials-13-02699-f001:**
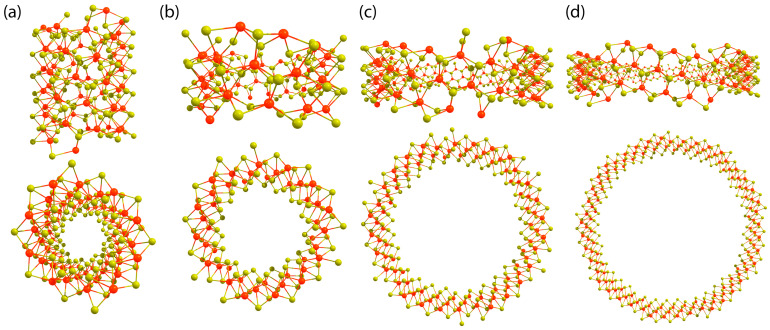
The side and top views of the considered nanotubes: (**a**) (8, 2), (**b**) (12, 3), (**c**) (24, 6), (**d**) (36, 9). In the figures, sulfur atoms are represented by yellow spheres, while tungsten atoms are represented by red spheres.

**Figure 2 nanomaterials-13-02699-f002:**
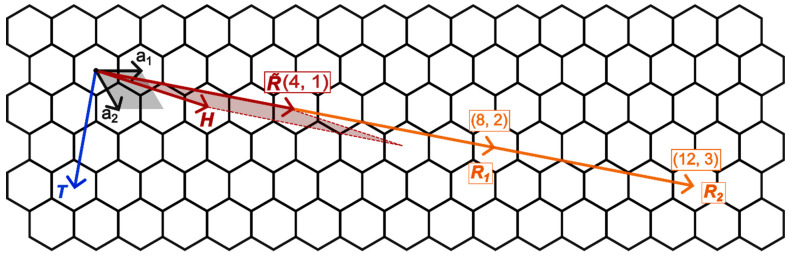
Chirality vectors of the nanotubes rolled up from a layer with hexagonal morphology.

**Figure 3 nanomaterials-13-02699-f003:**
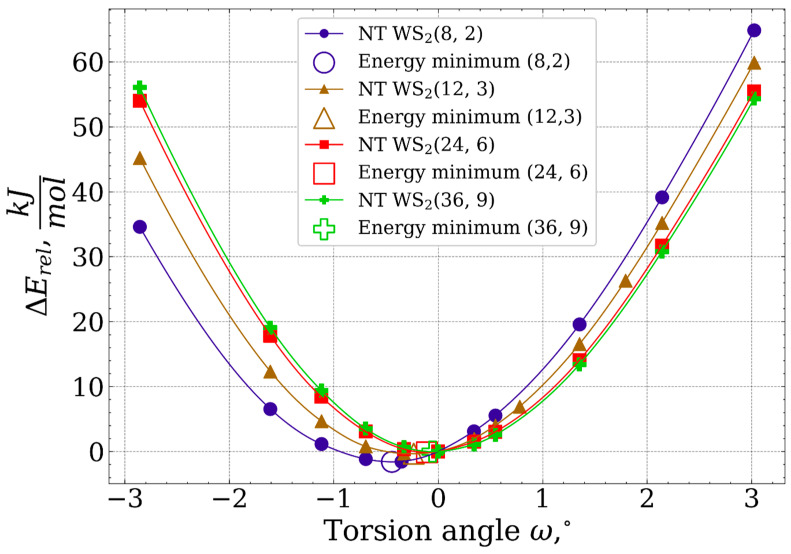
The dependence of the $\Delta E_{rel}$ on the torsion angle. $\Delta E_{rel}$ of structures with irrational Q are obtained here by interpolation [23] using SciPy [39] implementation of Akima interpolation [40]. Matplotlib [41] and Seaborn [42] were used for data visualizing.

**Figure 4 nanomaterials-13-02699-f004:**
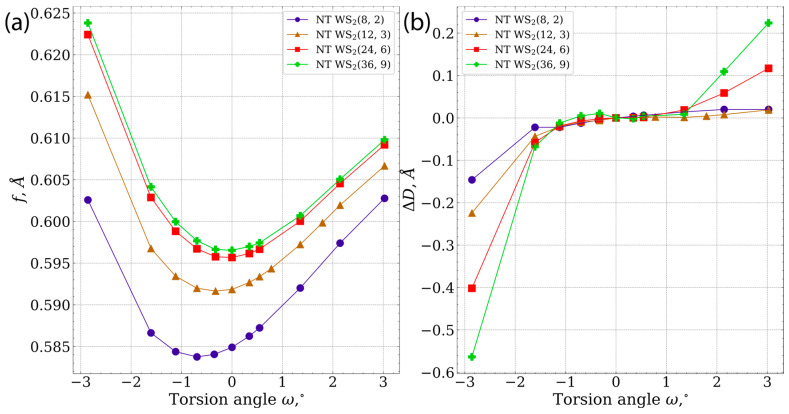
The torsion angle dependence on the values of (**a**) partial translation, (**b**) diameter changes.

**Figure 5 nanomaterials-13-02699-f005:**
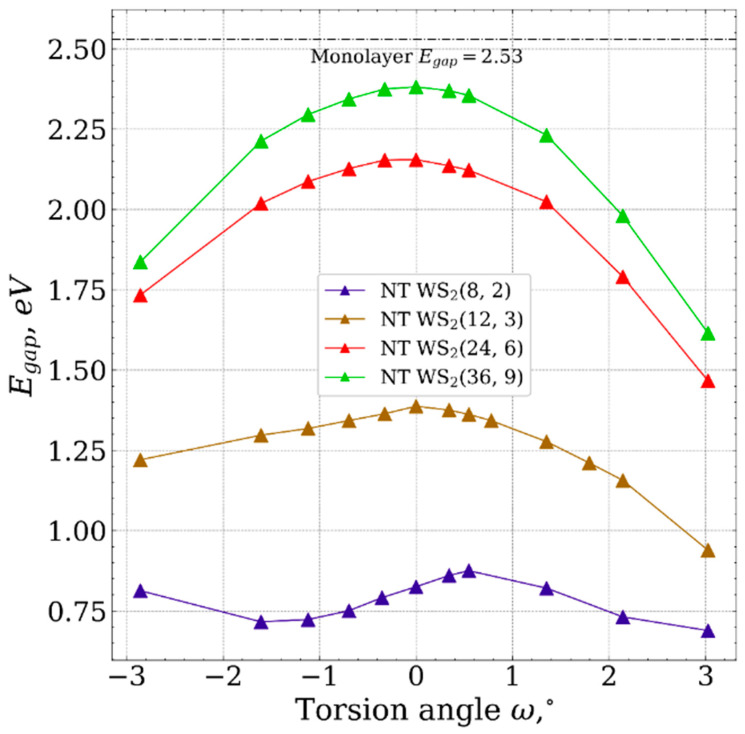
The dependencies of the values of band gap as the functions of torsion angle.

**Figure 6 nanomaterials-13-02699-f006:**
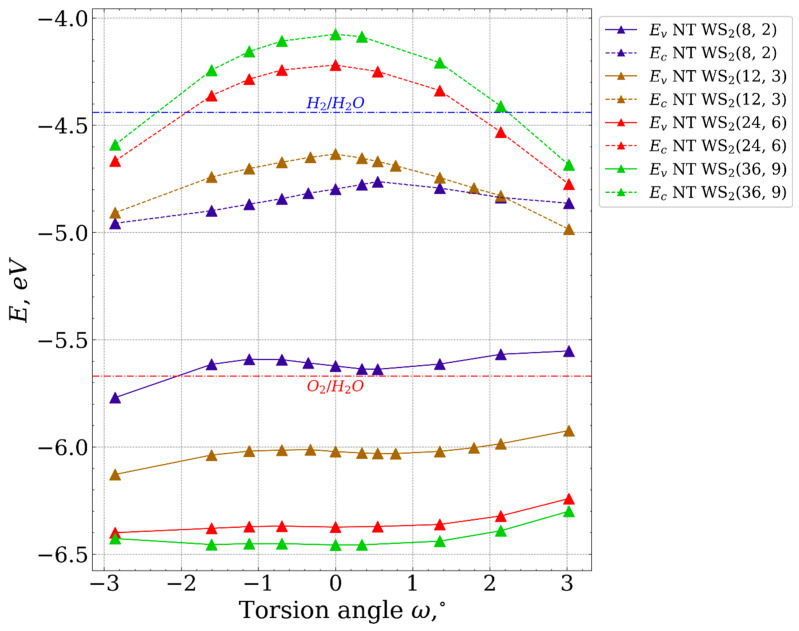
Energies of the top of the valence band and the bottom of the conduction band as the functions of torsion angle.

**Table 1 nanomaterials-13-02699-t001:** Comparison of calculated properties of bulk 2H-WS_2_ and monolayer with experimental data.

	2H-WS_2_ (*P6_3_/mmc*)		Monolayer	
	This Work	Experiment	This Work	Experiment
Lattice parameters, Å	*a* = 3.157, *c* = 12.530	*a* = 3.153 [34], *c* = 12.323 [34]	3.161	-
*E*_gap_, eV	1.63	1.4 [35]	2.53	2.4 [32,33]

**Table 2 nanomaterials-13-02699-t002:** The torsion angles for the nanotubes under investigation. The corresponding symmetry groups are provided for each torsion angle.

ω, °	(8, 2)	(12, 3)	(24, 6)	(36, 9)
−2.857	$L18_{8}\left(T_{18}^{7}C_{2}\right)$	$L27_{12}\left(T_{27}^{7}C_{3}\right)$	$L54_{24}\left(T_{54}^{7}C_{6}\right)$	$L81_{36}\left(T_{81}^{7}C_{9}\right)$
−1.607	$L64_{18}\left(T_{64}^{25}C_{2}\right)$	$L96_{27}\left(T_{96}^{25}C_{3}\right)$	$L192_{54}\left(T_{192}^{25}C_{6}\right)$	$L288_{81}\left(T_{288}^{25}C_{9}\right)$
−1.118	$L46_{18}\left(T_{46}^{18}C_{2}\right)$	$L69_{27}\left(T_{69}^{18}C_{3}\right)$	$L138_{54}\left(T_{138}^{18}C_{6}\right)$	$L207_{81}\left(T_{207}^{18}C_{9}\right)$
−0.695	$L74_{46}\left(T_{74}^{29}C_{2}\right)$	$L111_{69}\left(T_{111}^{29}C_{3}\right)$	$L222_{138}\left(T_{222}^{29}C_{6}\right)$	$L333_{207}\left(T_{333}^{29}C_{9}\right)$
−0.326	$L158_{130}\left(T_{158}^{62}C_{2}\right)$	$L237_{195}\left(T_{237}^{62}C_{3}\right)$	$L474_{390}\left(T_{474}^{62}C_{6}\right)$	$L711_{585}\left(T_{711}^{62}C_{9}\right)$
0.000	$L28_{18}\left(T_{28}^{11}C_{2}\right)$	$L42_{27}\left(T_{42}^{11}C_{3}\right)$	$L84_{54}\left(T_{84}^{11}C_{6}\right)$	$L126_{81}\left(T_{126}^{11}C_{9}\right)$
0.343	$L150_{28}\left(T_{150}^{59}C_{2}\right)$	$L225_{42}\left(T_{225}^{59}C_{3}\right)$	$L450_{84}\left(T_{450}^{59}C_{6}\right)$	$L675_{126}\left(T_{675}^{59}C_{9}\right)$
0.547	$L94_{28}\left(T_{94}^{37}C_{2}\right)$	$L141_{42}\left(T_{141}^{37}C_{3}\right)$	$L282_{84}\left(T_{282}^{37}C_{6}\right)$	$L423_{126}\left(T_{423}^{37}C_{9}\right)$
1.353	$L38_{28}\left(T_{38}^{15}C_{2}\right)$	$L57_{42}\left(T_{57}^{15}C_{3}\right)$	$L114_{84}\left(T_{114}^{15}C_{6}\right)$	$L171_{126}\left(T_{171}^{15}C_{9}\right)$
2.143	$L48_{38}\left(T_{48}^{19}C_{2}\right)$	$L72_{57}\left(T_{72}^{19}C_{3}\right)$	$L144_{114}\left(T_{144}^{19}C_{6}\right)$	$L216_{171}\left(T_{216}^{19}C_{9}\right)$
3.025	$L68_{58}\left(T_{68}^{27}C_{2}\right)$	$L102_{87}\left(T_{102}^{27}C_{3}\right)$	$L204_{174}\left(T_{204}^{27}C_{6}\right)$	$L306_{261}\left(T_{306}^{27}C_{9}\right)$

**Table 3 nanomaterials-13-02699-t003:** Results of calculations bulk crystal WS_2_, monolayer and chiral nanotubes $n\left({\tilde{n}}_{1},{\tilde{n}}_{2}\right)$, $\tilde{q}=14$, $q=n\tilde{q}$, t=14·f.

	Bulk	Mono	(8, 2)	(12, 3)	(24, 6)	(36, 9)
$Lq_{p}$	-	-	$L28_{18}$	$L42_{27}$	$L84_{54}$	$L126_{81}$
D, Å	-	-	9.22	13.82	21.66	41.49
ΔD,Å	-	-	1.46	1.07	0.60	0.42
t, Å	3.15	3.16	8.36	8.36	8.36	8.36
Δt, Å	-	0.01	0.17	0.07	0.03	0.01
*E*(f.u.), a.u, −88	0.297269	0.288675	0.220688	0.255304	0.279788	0.284615
*E_form_*(f.u.), kJ/mol	-	22.56	201.06	110.18	45.9	33.22
*E_str_*(f.u.), kJ/mol	-	-	178.5	87.62	23.33	10.66
*E_gap_*, eV	1.63	2.53	0.82	1.38	2.15	2.39
*E_v_*, eV	−3.96	−6.43	−5.62	−6.02	−6.37	−6.47
*E_c_*, eV	−2.33	−3.90	−4.80	−4.64	−4.22	−4.08

**Table 4 nanomaterials-13-02699-t004:** Properties of NTs structures corresponding to obtained energy minima.

	D ^a^*, Å	ω, °	$\Delta E_{rel}$, kJ/mol	$E_{gap}$^b^*, eV	G ^c^*, GPa
(8, 2)	10.67	−0.443	−1.601	0.79	39.16
(12, 3)	14.89	−0.234	−0.348	1.36	7.59
(24, 6)	28.26	−0.109	−0.139	2.15	0.84
(36, 9)	41.90	−0.053	−0.029	2.38	0.17

^a^—The diameter values are measured relate to the positions of the W atom centers. ^b^—Electronic band gap; ^c^—Shear module, calculation technique is described in works [22,23]; *—The values correspond to the results obtained from the quantum chemical calculation for the torsion angle that is closest to the energy minimum.

**Table 5 nanomaterials-13-02699-t005:** Band gap behavior of investigated nanotubes under torsional deformation.

ω, °	(8, 2)	(12, 3)	(24, 6)	(36, 9)
−2.857	Direct	Indirect	Indirect	Indirect
−1.607	Indirect	Indirect	Indirect	Indirect
−1.118	Indirect	Indirect	Indirect	Indirect
−0.695	Indirect	Indirect	Indirect	Indirect
−0.326	Indirect	Indirect	NA *	NA *
0.000	Indirect	Indirect	Indirect	Indirect
0.343	Indirect	Indirect	NA *	Indirect
0.547	Indirect	Indirect	Indirect	NA *
1.353	Indirect	Indirect	Indirect	Indirect
2.143	Indirect	Indirect	Indirect	Indirect
3.025	Indirect	Indirect	Indirect	Indirect

*—Cannot be obtained due to technical reasons.

## Data Availability

Data will be made available on request.

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
