# Peer review of "DFT Study of WS2-Based Nanotubes Electronic Properties under Torsion Deformations"

_nanomaterials, 2023, doi:10.3390/nano13192699_

Round 1

Reviewer 1 Report

In the manuscript “DFT study of WS2-based nanotubes electronic properties under torsion deformations”, the authors investigated the influence of torsional deformations on the electronic properties of WS2 nanotubes with chirality indices ?(4,1) using the density functional theory. This work has revealed that for nanotubes with smaller diameters, the structure obtained through rolling from a slab is not optimal and undergoes spontaneous deformation. In addition, this study demonstrated that the nanotube torsion deformation leads to a reduction in the bandgap. The results obtained are interesting and should be published. However, before publication, I hope you can make some revisions and deeper analysis. Specific comments are as follows:

1. In Figure 5, the values of band gap are asymmetric with respect to torsion angle w=0. What is the reason?

2. What is the effect of torsional deformations on the properties of photocatalytic devices? 

3. The conclusions section needs to be further condensed.

4. In order to enlarge the views of TMD-based nanotubes, the following reference may be helpful (New J. Phys., 24, 063012; Phys. Rev. B, 2023, 108, 045416). 

Reviewer 2 Report

The manuscript presents a comprehensive density functional theory (DFT) study into the electronic structures of WS2 nanotubes (NTs) with various chirality indices. The dependence of relative energy, band gap, shear moduli, and deformation on the torsion angle are explicitly illustrated and compared across different chiral WS2 NTs. The data presented in the manuscript holds substantial value for further studies, given that most of the current theoretical studies focus only on achiral WS2 NTs. I recommend the publication of the manuscript in Nanomaterials after minor revisions.

1.  The dependence of valence band maximum (Ev) and conduction band minimum (Ec) with the torsion angle in different WS2 NTs should also be elaborated and analyzed in the manuscript rather than only the band gap (Eg). The relative positions of Ev and Ec to the redox levels also have a decisive influence on the performance of WS2 NTs as water-splitting photocatalysts.

2. The band structure evolution with torsion angle in different WS2 NTs should be added as Supplementary Information. Understanding the relative positions of the valence and conduction band edges within the Brillouin zone is crucial for determining whether a WS2 nanotube exhibits a direct or indirect bandgap semiconductor behavior, which is another significant characteristic for photocatalytic applications.

3. I am particularly intrigued by the asymmetric dependence of all the properties under investigation on positive and negative torsion angles. The subtle chirality distinctions among the series of twisted WS2 nanotube structures likely play a pivotal role in this observation. It would be highly beneficial if the authors could offer additional chemical or structural insights into this phenomenon. However, considering the substantial amount of additional calculations and analysis required to address this aspect comprehensively, I can totally understand that this part of the investigation is currently not applied in this work.

There are some comments regarding the writing and the presentation of the manuscript:

1. Figure 1 is never referred to in the text. Please check. Additionally, please include the structure of the (8,2) WS2 NT in Figure 1.

2. line 95, “However, the investigation of the properties of nanotubes as a function of torsion angle ... ”

3. line 109, the expression “Strict accuracy criteria (10^-8 10^-8 10^-8 10^-8 10^-8 10^-18) ...” is confusing. Please verify if it is a typo or provide further clarification on the significance of these values in the context of Coulomb and exchange interactions.

4. The use of capitals in chapter headings is inconsistent throughout the manuscript. For example, “2.2 Application of Line Group Theory to Nanotubes with Hexagonal Morphology” and “2.3 Torsion deformation of NTs with hexagonal morphology”.

5. line 281, the statement is confusing. “Table 3 demonstrates that the forbidden energy gap (Eg), which is smallest for the bulk crystal, ...”. However, the Eg of (8,2) and (12,3) NTs is 0.82 eV, and 1.38 eV, respectively, both smaller than the Eg of bulk WS2 (1.63 eV). Why did the authors claim that the Eg of bulk crystal is the smallest?

6. line 223, “It is known that a visible-light-driven water-splitting catalyst needs to have Ev < -5.67 eV and Ec > -4.44 eV.”

7. Figure 3, the authors used the unit “kJ/mol” for E_rel, while “eV” has been used elsewhere in the manuscript. It is suggested to use "eV" consistently for energy units to maintain uniformity and facilitate comparisons between E_rel and E_form or E_rel and E_str.

8. The chapter Conclusion is missing a number index.

The English writing is already in good shape and doesn't require significant revision.
